# Progression of Watermelon Bud Necrosis Virus Infection in Its Vector, *Thrips palmi*

**DOI:** 10.3390/cells10020392

**Published:** 2021-02-14

**Authors:** Amalendu Ghosh, Bikash Mandal, Ralf G. Dietzgen

**Affiliations:** 1Insect Vector Laboratory, Advanced Centre for Plant Virology, ICAR-Indian Agricultural Research Institute, New Delhi 110012, India; preetykapoor01@gmail.com (P.S.); bmandal@iari.res.in (B.M.); 2Queensland Alliance for Agriculture and Food Innovation, The University of Queensland, St. Lucia, Brisbane, QLD 4072, Australia; r.dietzgen@uq.edu.au

**Keywords:** virus–vector relationships, thrips, orthotospovirus, localization, WBNV, melon thrips

## Abstract

Thrips are important pests of agricultural, horticultural, and forest crops worldwide. In addition to direct damages caused by feeding, several thrips species can transmit diverse tospoviruses. The present understanding of thrips–tospovirus relationships is largely based on studies of tomato spotted wilt virus (TSWV) and Western flower thrips (*Frankliniella occidentalis*). Little is known about other predominant tospoviruses and their thrips vectors. In this study, we report the progression of watermelon bud necrosis virus (WBNV) infection in its vector, melon thrips (*Thrips palmi*). Virus infection was visualized in different life stages of thrips using WBNV-nucleocapsid protein antibodies detected with FITC-conjugated secondary antibodies. The anterior midgut was the first to be infected with WBNV in the first instar larvae. The midgut of *T. palmi* was connected to the principal salivary glands (PSG) via ligaments and the tubular salivary glands (TSG). The infection progressed to the PSG primarily through the connecting ligaments during early larval instars. The TSG may also have an ancillary role in disseminating WBNV from the midgut to PSG in older instars of *T. palmi*. Infection of WBNV was also spread to the Malpighian tubules, hindgut, and posterior portion of the foregut during the adult stage. Maximum virus-specific fluorescence in the anterior midgut and PSG indicated the primary sites for WBNV replication. These findings will help to better understand the thrips–tospovirus molecular relationships and identify novel potential targets for their management. To our knowledge, this is the first report of the WBNV dissemination path in its vector, *T. palmi*.

## 1. Introduction

Thrips belong to the order Thysanoptera, which includes more than 7700 species of tiny, fringed winged insects [1]. Globally, thrips are considered the most economically damaging pests of a wide range of food, feed, and fiber crops [2]. They cause extensive feeding damage by piercing and sucking the epidermal and mesophyll cells, resulting in silvery scars on the leaves, stems, flowers, and fruits and sometimes induce galls. In addition to being plant pests, some thrips can also spread several tospoviruses (genus *Orthotospovirus*, family *Tospoviridae*, order *Bunyavirales*) that cause significant economic losses to crop production across the globe [3]. High fecundity, short lifecycle, high mobility, preference for concealed spaces, and wide host range make them efficient virus vectors as well as pests. To date, 16 thrips species are known to transmit 29 tospoviruses [4,5,6]. The present understanding of thrips–tospovirus interactions is largely based on Western flower thrips, *Frankliniella occidentalis* (Thysanoptera: Thripidae) and tomato spotted wilt virus (TSWV). Based on this model system, it is understood that thrips transmit tospoviruses in a persistent, propagative manner, and adults are the primary transmitters, as they bear functional wings to fly long distances. Adult thrips become viruliferous if virus acquisition takes place in an early larval instar [7,8]. TSWV replicates in the midgut and surrounding tissues and reaches salivary glands through either the tubular salivary glands (TSG) or thread-like connecting ligaments [7,9,10]. Although many other circulative viruses are thought to travel from the gut to the salivary glands via a hemolymph route, tospovirus particles have not been encountered in the hemolymph in any study [4]. It is thought that the physical contact between the salivary glands and midgut in *F. occidentalis* is disrupted as the size of the head capsule increases in the following instars [11]. *F. occidentalis* cannot become viruliferous if virus acquisition takes place after the physical disconnection of the midgut from the salivary glands during older instars [4]. Little is known about whether other predominant tospoviruses follow the same route of dissemination in their respective thrips vectors. 

Melon thrips (*Thrips palmi*, Thysanoptera: Thripidae) is a key pest of vegetables, legumes, and ornamentals. The distribution of *T. palmi* was thought to be restricted to southern Asia, but it has spread throughout Asia in recent decades. *T. palmi* has widely invaded the Pacific, Florida, the Caribbean, South America, Africa, and Australia [12,13,14,15,16,17,18,19,20]. It is now considered the predominant tospovirus vector in Asia [12]. To date, seven tospoviruses are known to be transmitted by *T. palmi* viz. groundnut bud necrosis virus (GBNV), melon yellow spot virus (MYSV), calla lily chlorotic spot virus (CCSV), watermelon silver mottle virus (WSMV), watermelon bud necrosis virus (WBNV), tomato necrotic ringspot virus (TNRV), and capsicum chlorosis virus (CaCV) [4,5]. WBNV is among the predominant tospoviruses in Asia, causing yield losses up to 100% [21,22,23]. The relationship of *T. palmi* with WBNV is less explored. WBNV is transmitted by *T. palmi* in a persistent propagative manner. Exposure to WBNV negatively affects the adult life span, fecundity, and survival of *T. palmi* [8]. The present study reports the localization of WBNV nucleocapsid (N) protein in different life stages of *T. palmi* after the virus is acquired by early larval instar and speculates as to the progression of WBNV infection in its vector. 

## 2. Materials and Methods

### 2.1. Developing a Homogeneous Population of Thrips palmi 

A homogeneous population of *T. palmi* was developed from a single adult female collected from the stock culture maintained at Advanced Centre for Plant Virology, Indian Agricultural Research Institute (IARI), New Delhi. Identification of the thrips was based on established morphological keys [13,24] and on sequencing mitochondrial oxidase subunit I (mtCOI) [25]. The population was maintained on healthy eggplants (*Solanum melongena* var. Navkiran) under controlled conditions at 28 ± 1 °C, 60 ± 10% relative humidity, and 8 h dark/16 h light. 

### 2.2. Establishing Watermelon Bud Necrosis Virus Culture

The initial inoculum of WBNV was collected from a pure culture maintained in cowpea at the containment facility of Advanced Centre for Plant Virology, IARI. Healthy cowpea plants (*Vigna unguiculata* var. Pusa Komal) were raised from seeds in an insect-proof plant growth chamber. Cowpea plants at 2-leaf stage were sap-inoculated following a protocol described by Mandal and colleagues [26] with some modifications. Briefly, sap was extracted from WBNV-infected cowpea leaves in ice-cold 0.01 M phosphate buffer (pH 7) containing 0.2% freshly added 2-mercaptoethanol with a tissue to buffer ratio of 1:4 wt/vol. A pinch of Celite was dusted on the leaves of cowpea plants, and inoculum was applied on leaf surfaces by gently rubbing with gloved hands. The inoculated plants were left for 5 min, washed with distilled water, and maintained at 24 ± 1 °C under insect-proof conditions. The plants were regularly monitored for symptom appearance. All the plants were tested by reverse transcription-PCR (RT-PCR) using WBNV-specific primers 14 days post inoculation (dpi) as described below. 

### 2.3. Generating a Viruliferous Thrips palmi Population

Eggs of *T. palmi* were collected from an artificial oviposition setup as described by Jangra and colleagues [27]. The eggs were placed on a moistened tissue paper and incubated at 28 °C. The freshly emerged first instar larvae (L1) were collected with the help of a Camel hairbrush and used for WBNV acquisition. For the virus acquisition setup, a Thermocol (a form of polystyrene) block (2 × 2 × 5 cm^3^) was glued to the bottom of a plastic box (9 cm diameter, 8 cm height), and a WBNV-infected cowpea leaf was placed on it with a needle-point paper pin (2 cm). The leaf started floating as the box was filled with water up to 6.5 cm in height. The paper pin allowed the leaf to float on water at the center of the box. About 20 *T. palmi* L1 were released on WBNV-infected floating cowpea leaf for a 12 h acquisition access period (AAP) at 28 ± 1 °C, 60 ± 10% relative humidity. The L1 were confined to the leaf surface due to water periphery and forced to feed on infected leaves only. Following the AAP, thrips were transferred to healthy eggplant and reared to adult stage under insect-proof controlled conditions. *T. palmi* were randomly collected from the WBNV-exposed and non-exposed populations and tested by RT-PCR to confirm virus-infection. Different life stages of WBNV-exposed and non-exposed *T. palmi* were collected for the virus localization study. 

### 2.4. Confirming WBNV Infection by RT-PCR

RT-PCR was performed to confirm the infection of WBNV in cowpea plants and *T. palmi*. Total RNA was extracted from cowpea leaves 14 dpi using a RNeasy Plant Mini Kit (Qiagen, Germantown, MD, USA) following the manufacturer’s protocol. *T. palmi* RNA was extracted using NucleoSpin RNA XS (Macherey-Nagel, Dueren, Germany) following the manufacturer’s protocol. Total RNA was eluted in 25 µL RNase-free water. Complementary DNA was synthesized using a Verso cDNA synthesis Kit (Thermo Fischer Scientific, Waltham, MA, USA) with an RNA template, and WBNV-specific primers Wb-F and CaGbWb-R [28]. PCR was carried out in 25 µL reactions containing 20–30 ng cDNA, 2.5 µL 10X PCR buffer (Thermo Fischer Scientific), 0.4 µM each forward and reverse primers, 260 µM dNTP mix (Thermo Fischer Scientific), and 2 U DreamTaq polymerase (Thermo Fisher Scientific). PCR was performed in a T100 Thermal Cycler (Bio-Rad, Hercules, CA, USA) with 94 °C for 5 min followed by 30 cycles of 94 °C for 30 s, 56 °C for 90 s, and 72 °C for 1 min followed by a final extension at 72 °C for 10 min. PCR products were resolved on 1% agarose gel stained with GoodView™ (BR Biochem, Delhi, India) and observed in a gel documentation system (MasteroGen Inc., Hsinchu, Taiwan). 

Hemolymph from a group of WBNV-exposed and non-exposed *T. palmi* of different life stages was collected and tested by RT-PCR as described above for the presence of WBNV. 

### 2.5. Dissection and Immunolabeling of Alimentary Canal and Salivary Glands of Thrips palmi

L1, second instar larva (L2), prepupa (P1), pupa (P2), and adult thrips were placed on glass slides, and 20–30 µL of ice-cold phosphate-buffered saline (PBS, pH 7.4) was added to the specimens. Individual thrips were dissected with the help of “000” entomological pins and a surgical blade. The alimentary canal and salivary glands were carefully removed from rest of the thrips’ body and placed on a clean glass slide. 

The specimens were allowed to dry at room temperature and fixed with 4% paraformaldehyde in 50 mM sodium phosphate (pH 7.0) for 1 h at room temperature and immunolabeled [9]. The slides were rinsed thrice with PBS and left overnight in PBS containing 1% Tween 20 at 4 °C. All incubation steps were performed in a parafilm-sealed box to prevent desiccation of specimens. After 18 h, the slides were washed three times with PBS and incubated with blocking buffer (PBS, 0.1% Tween 20, and 5% non-fat dry milk (NFDM)) for 30 min at room temperature. The specimens were incubated with rabbit antibodies against WBNV N protein diluted in PBS + 1% NFDM at a concentration of 2 µg/µL for 2 h 30 min. Specimens were washed with PBS thrice and incubated for 2 h in the dark with 0.02 µg/µL fluorescein isothiocyanate (FITC)-conjugated goat anti-rabbit secondary antibody (Thermo Fischer Scientific) diluted in PBS + 1% NFDM. The slides were washed thrice again with PBS and incubated for 2 h with phalloidin conjugated to Alexa Fluor 594 (Thermo Fischer Scientific) diluted at 4 units/mL. Slides were washed with PBS and distilled water three times each. The incubation chamber was removed, and specimens were air-dried at room temperature. One drop of 50% glycerol was added to the specimen, covered with a glass slide, and sealed with nail polish. The slides were stored at 4 °C in the dark. 

### 2.6. Localization of WBNV N Protein by Confocal Laser Scanning Microscopy

The slides with immunolabeled gut and salivary glands of each life stage of *T. palmi* were examined with a confocal laser scanning microscope (TCS SP5 II, Leica, Wetzlar, Germany). A 488 nm Argon laser diode was used for FITC with a bandpass (BP) filter of 490–525 nm, and a diode-pumped solid-state (DPSS) 561 nm laser was used for Alexa Fluor 594 with a BP filter of 590–617 nm. The gain was fixed at 100 for FITC and 800 for Alexa Fluor 594 to enable comparison of fluorescence signals across all images. Likewise, pinhole aperture was set for FITC at 100 and 62 µm for Alexa Fluor 594. To acquire the images, a 10x-dry objective was used with a variable zoom scale. The fluorescence signals of FITC and Alexa Fluor 594 for a specimen were captured simultaneously at the same depth, focus, and magnification and merged using LAS AF software (Leica). The bright-field images were captured in an EA prime inverted microscope (Leedz Microimaging Ltd., Leeds, UK) and processed using LMI Image analysis software. 

### 2.7. Quantification of the Fluorescence Signal

The FITC-specific fluorescence indicative of N protein localization was measured in different parts of the WBNV-exposed and non-exposed specimens by ImageJ to estimate the level of virus infection [9,29]. The mean intensity of the fluorescence signal (green channel, FITC) was calculated as raw integral density in the sum of pixel value divided by the area with fluorescence signal and expressed as the mean grey value (MGV). The automatic scale was muted to consider the number of pixels as the unit of measurement. The different parts of the alimentary canal, such as the foregut, midgut, hindgut, and Malpighian tubules, ligaments, and salivary glands, were selected using a freehand tool, and the percent of area with fluorescence and MGV were measured. The length and diameter of the foregut, midgut, hindgut, Malpighian tubules, ligaments, and salivary glands were measured after calibrating each image by transforming the pixel value into micrometers (µm). 

## 3. Results

### 3.1. Thrips palmi Population

The homogenous thrips population developed from a single adult female was identified as *T. palmi* based on the morphological keys and DNA sequence. The adults of *T. palmi* were yellow. The head was quadrangular in shape with seven segmented antennae. Three brick-red ocelli were visible on the top of the head in a triangular formation. A pair of interocellar setae originated outside the ocellar triangle. A black line appeared on the dorsal side of the body at juncture of the wings. Female adults were identified by the presence of a sharp ovipositor at the apex of the abdomen, whereas males had a bluntly rounded apex of the abdomen. Females were comparatively larger than males. Further, the nucleotide sequence of mtCOI confirmed the identity as *T. palmi*. The 653 bp sequence of mtCOI was 100% identical to that of other *T. palmi* isolates (MN594549, MW020346) available in GenBank. The sequence can be retrieved with accession number MT992047. 

### 3.2. Watermelon Bud Necrosis Virus Culture

WBNV symptoms were observed on leaves of inoculated cowpea plants 10–14 dpi. WBNV-infected plants showed numerous chlorotic spots and yellowing of leaves, necrotic streaks on stems, stunted growth, and deformed fruits. RT-PCR with WBNV-specific primers amplified a distinct band of 550 bp. Further, the nucleotide sequence of the amplified product (accession number: MW080513) was more than 99% identical to sequences of other WBNV isolates (EU216028, MF805799) in GenBank.

### 3.3. Developmental Stages and Viruliferous Population of Thrips palmi 

The adult females of *T. palmi* laid their eggs inside leaf tissue. Intact eggs were collected in the artificial oviposition setup described in the materials and methods. Freshly laid eggs were translucent and turned yellowish on maturation. Eggs were kidney-shaped and ~300 µm long. Eggs were hatched after 85–90 h when incubated at 28 °C. Freshly emerged larvae (L1) were translucent and ~400 µm long. Immediately after hatching, L1 started feeding on leaves. A median green line was visible along the length of the body, probably the alimentary canal. After 24–36 h, L1 molted into L2. L2 were bigger, yellow, and active feeders. The developmental period of L2 was about 72 h. Larvae did not bear wing pads and had a smaller number of segments in the antennae. Both P1 and P2 pupal stages were with wing pads, sessile, and non-feeding. The P1 stage lasted for approximately 48 h. The adults emerged from P2 after 60–72 h. Adult *T. palmi* had fully developed wings and were highly mobile, actively feeding preferably on the lower surfaces of leaves. The WBNV-exposed *T. palmi* population suffered a lower survival during the emergence of adults from P2 than the non-exposed population (data not shown). In RT-PCR, a 550 bp WBNV-specific amplicon confirmed that the *T. palmi* population was infected with WBNV (data not shown). 

### 3.4. Localization of WBNV in Alimentary Canal and Salivary Glands of Thrips palmi during Different Developmental Stages

The alimentary canal originated from the buccal cavity and ran medially along the length of the thrips body. It could be divided into foregut, midgut, and hindgut sections (Figure 1). The alimentary canal was white, but sometimes the midgut appeared greenish due to the presence of plant sap. The anterior foregut branched into paired translucent lobes, the salivary glands. Freshly emerged L1 were exposed to WBNV, and the virus was immunolocalized in the alimentary canal and salivary glands at different life stages using N protein antibody tagged with FITC-conjugated secondary antibody. When viewed in a confocal microscope, Alexa Fluor 594 phalloidin that has a high-affinity for F-actin yielded red fluorescence at excitation/emission of 581/609 nm, while FITC-conjugated secondary antibody was characterized by bright green fluorescence at excitation/emission of 490/525 nm. The presence of WBNV was identified by FITC-specific green fluorescence. The merging of both phalloidin and FITC fluorescence signals yielded yellow to orange shades in the image. The level of fluorescence varied between biological replicates irrespective of the life stages. The appearance and arrangement of the alimentary canal and salivary glands were similar in all the instars of *T. palmi* but increased in size in older instars. The ligaments and internal organs of *T. palmi* were very soft and fragile. Some tissues were ruptured or disintegrated in the specimens during processing and immunolabeling. A total of 103 specimens were dissected during the study and summary observations are presented here. The detailed observations for each instar are described below.

#### 3.4.1. Localization of WBNV in L1 Stage

The foregut after originating from the buccal cavity led into the esophagus that further opened into the midgut lumen. In L1, the foregut was very short, around ~180 µm. It was narrow throughout its length with a maximum expansion of 5.83 µm. The midgut of *T. palmi* constituted the major portion of the alimentary canal. It was divided into three parts with a constriction between midgut 1 (MG1) and midgut 2 (MG2). The distal part of MG2 was tubular and extended into midgut 3 (MG3) at its posterior. MG2 formed a coiled structure where the posterior end touched MG1. This midgut loop was uncoiled during processing and washing in the experimental specimens. The total length of the midgut was around 1250 µm. MG1 was broader with a diameter of 75 µm. The anterior portion of MG2 was around 35 µm in diameter, whereas the tubular portion had a diameter of about 10 µm. MG3 resembled a wide globular chamber with a maximum expansion of 49.4 µm. MG3 sometimes appeared dark green. The midgut led into the small tubular hindgut that ended in an anal opening. The hindgut length was about 351 µm with a diameter of around 30 µm. Two pairs of Malpighian tubules arose from the junction of the midgut and hindgut. The average length of the Malpighian tubules was 300 µm with ~10 µm diameter. 

Salivary glands had two parts, the principal salivary glands (PSG) and TSG. The PSG appeared as a paired ovoid structure that was connected through a small piece of tissue. The PSG was placed between the head and prothorax in larvae. A pair of ligaments connected the basal region of the PSG to the anterior midgut 1 (Figure 2). The TSG emerged from the basal regions of the gland, led down the alimentary canal, and touched at the junction of MG1 and MG2. The TSG was about 372 µm long and 3.4 µm in diameter. 

After acquisition of WBNV, the first FITC fluorescence signal indicating the localization of N protein was observed prominently in the anterior midgut comprising MG1 and MG2 8 h post WBNV acquisition (Figure 3). The intensity of fluorescence measured in MGV was 26.8. However, the tubular portion of MG2 showed no fluorescence, indicating no WBNV presence. Comparatively lower intensity of green fluorescence was observed in MG3 (10.34 MGV). Infection of epithelial-like cells in MG1 was evident from the presence of green fluorescence 12 h post acquisition (Figure 4). A distinct but low intensity green fluorescence was detected in the PSG, and thread-like ligaments connecting the midgut to the PSG (Figure 4) 12 h post acquisition indicated the dissemination of WBNV from the anterior midgut to the PSG via ligaments during the L1 stage. The TSG was not infected by WBNV, indicated by the absence of green fluorescence (Figure 4). However, the accumulation of WBNV was very low in the PSG during the L1 stage. A low level of green fluorescence was also visible in the foregut (6.7 MGV). On the other hand, no green fluorescence was observed in the hindgut or Malpighian tubules. These results indicate that the anterior portion of the midgut was the first to be infected with WBNV during the L1 stage. The remaining portion of the midgut loop was not infected in the early larval instar, indicated by the absence of FITC fluorescence. WBNV infection was comparatively low in MG3, despite its proximity to the anterior midgut through a loop-like structure. Accumulation of WBNV in ligaments during the early L1 stage indicates their role in the dissemination of the virus to the PSG.

#### 3.4.2. Localization of WBNV in L2 Stage

The appearance of the alimentary canal in L2 was similar to L1. The foregut was small and narrow. The average length of foregut was around 382 µm. The midgut constituted the major portion of the alimentary canal. The total stretch (1904 µm) of the midgut was slightly longer than in L1. The constriction between MG1 and MG2 was more prominent in L2. MG1 was broader with a diameter of 94 µm. The anterior portion of MG2 was around 67.5 µm in diameter followed by a tubular portion of ~ 30 µm in diameter. MG3 was globular shaped with an average width of 61.7 µm. In many specimens, the midgut appeared to be collapsed when not filled with food materials. The ligaments connected the PSG and the midgut in a similar fashion as in L1. The TSG was found to be attached to the midgut. The Malpighian tubules were approximately 323 µm long with a mean diameter of 11.2 µm.

Localization of WBNV N protein in the midgut and hindgut of L2 was distinguished by FITC green fluorescence 48–60 h post acquisition by L1. Green fluorescence was prominent in MG1 and the anterior portion of MG2 with an MGV of 28.1 (Figure 3). Fluorescence increased in the anterior portion of MG3 (29.95) compared to L1. The intensity of green fluorescence also increased in ligaments and the PSG. The posterior region of MG3 showed no fluorescence in some specimens, while MGV was around 18 in the rest of the specimens. These results indicate that during the L2 stage, virus titer increased in the anterior portion of the midgut compared to L1. Green fluorescence, indicating WBNV N protein, increased in the hindgut (17.9 MGV). A low intensity fluorescence was also observed in the foregut. The dissemination of virus infection from the anterior midgut to ligaments and the PSG was prominent in L2 60 h post acquisition (Figure 5). WBNV infection in the TSG was evident from FITC fluorescence in some specimens (Figure 5), whereas it was absent in others during the L2 stage. 

#### 3.4.3. Localization of WBNV in Pupal Stages

There were no major changes in the appearances of the alimentary canal and salivary glands during the pupal stages. However, the alimentary canal grew longer. In most of the dissections, the alimentary canal of the pupal specimens had a narrow diameter or shrunk compared to L2, which might be due to less or no intake of food during these stages. The connection of the PSG with the midgut via ligaments remained intact during P1 and P2 stages. The TSG was found to be touching the midgut like in earlier instars. The total length of foregut was about 424 µm, whereas the midgut was 1816 µm, and the hindgut was 460 µm long. The diameters of MG1, MG2, and MG3 were 50, 37, and 9.5 µm, respectively. The posterior portion of MG2 did not form a tubular structure. Malpighian tubules were also visible in pupal specimens, with an average length of about 522 µm and a diameter of about 12 µm. 

Localization of WBNV N protein in the P1 stage was distinguished by FITC green fluorescence 5 days post acquisition by L1. The green fluorescence increased up to 55.5 MGV in the anterior portion of the midgut, indicating the replication of WBNV in this part of the gut during P1 (Figure 3). Stronger green fluorescence was observed in the PSG than in the preceding instars, probably due to the replication of WBNV in salivary glands. The mean intensity of green fluorescence in the PSG was 23.11 MGV. An increased level of FITC fluorescence was observed in connecting ligaments. A corresponding (13.3 MGV) fluorescence in the hindgut suggests its role in virus replication. 

In P2, the arrangement of alimentary canal structures was similar to P1. Strong intensity of green fluorescence was observed in the anterior midgut and PSG at 8 days post WBNV acquisition. Ligaments were found to be connected to the basal part of MG1. Strong FITC fluorescence in the ligaments indicated the accumulation of WBNV (Figure 6). The TSG was connected to the midgut and was infected by WBNV, indicated by the FITC green fluorescence (Figure 7). WBNV infection was also detected in the hindgut and Malpighian tubules during the P2 stage (Figure 8). 

#### 3.4.4. Localization of WBNV in Adult Stage

In *T. palmi* adults, both the alimentary canal and salivary glands were bigger than in the preceding instars. The foregut was 550.5 µm long with a diameter of 42 µm. The midgut measured about 1967.5 µm. The diameter of MG1 was about 89 µm in fed specimens and 38 µm in starved specimens. Constriction between MG1 and MG2 was prominent in adult specimens. MG2 was flattened up to 88 µm, but only up to 37 µm in starved conditions. The posterior MG2 did not form a tubular structure. MG3 had a globular shape with an average diameter of 110.5 µm, but only 35 µm in starved adults. The PSG was placed in the thoracic segments of the adults. The ligaments connecting the PSG with the anterior midgut were still intact but appeared shorter compared to the larval stages (Figure 2B). The attachment of the TSG to the midgut was still visible. Malpighian tubules at the junction of the midgut and hindgut were 568 µm long with a diameter of 15 µm. The hindgut was about 467.5 µ long with a 65.1 µm diameter.

Localization of WBNV N protein was visualized in adult *T. palmi* at 12 days post acquisition of the virus by L1. WBNV infection spread almost throughout the entire gut, indicated by FITC green fluorescence. Prominent green fluorescence was observed in the midgut, hindgut, Malpighian tubules, and PSG (Figure 3). In some specimens, FITC fluorescence was detected in the posterior section of the foregut close to the junction with the midgut. However, green fluorescence in other sections of the foregut was negligible (Figure 9). Significantly higher green fluorescence signals indicating WBNV accumulation were recorded in MG1 and MG2 (80.44 MGV) than in any other life stage (Figure 3). Green fluorescence was 46 MGV in MG3. In the PSG, green fluorescence (52.6 MGV) was highest during the adult stage. The peak level of green fluorescence in the midgut and PSG indicated the highest virus replication in these tissues during the adult stage. Comparatively less fluorescence was observed in Malpighian tubules (29.42 MGV). Infection of circular and longitudinal muscle tissues lining the midgut was evident by the distinct fluorescent lattice pattern (Figure 9), but, in some cases, was restricted to a few rows of longitudinal muscle cells. However, epithelial-like cells and their infection were inconspicuous in WBNV-exposed adult *T. palmi*. 

No corresponding green fluorescence signal was observed in the respective life stages of aviruliferous thrips (Figure 3). 

No WBNV-specific amplification was observed in RT-PCR with hemolymph of viruliferous *T. palmi* indicative of absence of WBNV from the hemolymph. 

The development of *T. palmi*, from egg to adult, indicating infection and gradual dissemination of WBNV at each life stage as revealed by our studies was compiled in a schematic diagram (Figure 10) to summarize the progression of WBNV infection during thrips development. 

## 4. Discussion

Tospoviruses are vectored by at least sixteen different species of thrips [4,5,6]. The *F. occidentalis*-TSWV model pathosystem has been studied widely to describe the relationship of thrips and tospoviruses. The localization of TSWV in its vectors was reported by Nagata et al. [7], Montero-Astúa et al. [9], de Assis Filho et al. [30], Han et al. [10], and Tsuda et al. [31]. It is understood that adult thrips can only transmit tospoviruses if acquisition takes place during the early larval stages. In the case of TSWV, initial replication occurs in epithelial cells of MG1, and infection subsequently spreads to surrounding muscle cells, leading to further infection of the entire gut of *F. occidentalis*. TSWV infection progresses to the PSG through the TSG or connecting ligaments [7,9,10]. Little is known about whether other tospoviruses follow the same route of infection in their respective thrips vectors. The present study reports for the first time the progression of WBNV infection in the developmental stages of its vector, *T. palmi* by serological localization of the viral N protein. 

*T. palmi*-transmitted WBNV is a predominant tospovirus that infects plants in the families Cucurbitaceae, Solanaceae, Malvaceae, Euphorbiaceae, and Asteraceae. WBNV can cause up to 100% yield loss, and epidemics have forced many farmers to relinquish their crops in India [23]. Exposure to WBNV negatively affects the fitness and development of its vector, *T. palmi* [8,32]. Adult life span, fecundity, and survival of WBNV-exposed *T. palmi* was reduced post virus acquisition by the L1 stage. In the present study, *T. palmi* was exposed to WBNV during L1 and carried through the developmental stages (Figure 10). The progression of infection in different stages of development was visualized using an FITC fluorescent goat anti-rabbit secondary antibody tagging WBNV-specific antibodies directed against the viral N protein. The initial infection of WBNV was located in the anterior region of the midgut comprising MG1 and MG2 immediately after virus acquisition by L1. Infection in epithelial-like cells of MG1 was evident 12 h post acquisition. Epithelial-like cells appear to be involved in the replication of WBNV in early larvae. An in-depth study of the cell type(s) involved would be necessary to shed further light on the replication and cellular movement of this virus in different developmental stages of thrips. In the case of TSWV and *F. occidentalis*, infection starts in epithelial cells of MG1 [9,30,33]. Involvement of MG2 in the initial infection of TSWV in *F. occidentalis* is not clear. Replication of WBNV started in MG1 and MG2 of *T. palmi*, indicated by increased virus-specific fluorescence and spread through the midgut during the L2 stage. We do not have any evidence to suggest that different regions of the midgut may have distinct chemistries, types of cells, and/or functions in *T. palmi* that affect the level of WBNV infection, but such speculation may be worth further investigation. It is interesting to note that Bandla and colleagues [34] detected a putative TSWV cellular receptor in MG1, 2, and 3 in *F. occidentalis* larvae. In other insect systems, such differences have been observed where midgut regions have distinct morphological, histological, and genetic properties [35,36,37] that result in different regions being variably competent to virus infection. The infection and replication of WBNV in the anterior midgut may also be supported by factors such as pH and ion concentration. 

The primary infection of WBNV at the anterior midgut in L1 disseminated to the PSG in later hours. The PSG of *T. palmi* appeared to be connected to the anterior midgut by ligaments. Being tubular in shape, the TSG led down and touched the alimentary canal at the conjunction of MG1 and MG2. In *F. occidentalis*, ligament-like structures connecting the PSG to the anterior MG1 have been proposed as the route of TSWV dissemination from the midgut to the PSG [7,30]. In contrast, Montero-Astúa and colleagues [9] did not observe TSWV infection in ligaments during any stage of *F. occidentalis*, and these authors proposed that the TSG serves as a channel of tospovirus movement from the midgut to PSG. In the present study, infection of WBNV initially reached the PSG through the connecting ligaments during the late L1 stage visualized at 12 h post acquisition. However, the titer of WBNV was very low in the PSG during this phase. WBNV could not be localized in the TSG at this time. In a few specimens, WBNV infection in the TSG was first observed during the L2 stage 60 h post acquisition, while it was prominent during pupal stages in the rest of the specimens. It was evident in the present study that the infection spread to the PSG of *T. palmi* before it was detected in the TSG. It is unclear whether the TSG has an ancillary role in the dissemination of the virus from the anterior midgut to the PSG in older larvae and pupae or TSG infection is a result of virus movement from the PSG to TSG. The ligaments are likely important in virus infection, as muscle cells can retain the virus across the trans-stadial passage unlike epithelial cells [7,9,30]. However, larvae are inefficient in virus transmission, which may be due to low virus titer. Significant replication of WBNV in the PSG was recorded during the pupal stages. Since pupae are sessile and non-feeding, they cannot transmit the virus, despite the high titer in the PSG. 

In adult *T. palmi*, WBNV infection levels of the PSG were much higher than in the preceding instars, supporting its ability for virus transmission to host plants. The strongest virus-specific fluorescence in the midgut and PSG indicated the primary replication sites of WBNV in *T. palmi*. In fact, the intensity of WBNV-specific fluorescence was the highest in the anterior midgut throughout all the life stages of *T. palmi.* These findings are consistent with earlier reports by Han and colleagues [10], where 1.5 times more virus infection was recorded in MG1 than in salivary glands in adults of soybean thrips, *Neohydatothrips variabilis* (Beach). In contrast, Montero-Astúa and colleagues [9] reported less infection of TSWV in MG1 than in the PSG in adults of *F. occidentalis*. The observed difference may be due to the studied virus and biophysiological characters of thrips species. Infection of WBNV had spread almost through the entire gut of viruliferous adults 12 days post acquisition. Infection in longitudinal and circular muscle tissues of the midgut was more intense during the adult stage. Similar patterns of infection in visceral circular and longitudinal muscle tissues of *F. occidentalis* and *N. variabilis* by tospoviruses were reported by Nagata, Han, and colleagues [10,38]. The hindgut, Malpighian tubules, and TSG may also serve as replication sites of WBNV in *T. palmi* adults. Localization of WBNV in the posterior section of the foregut of adult thrips is consistent with previous reports [7,9,30]. We hypothesize that WBNV reaches the foregut by lateral movement of the virus from anterior midgut cells. 

Hemolymph of WBNV-exposed *T. palmi* was tested by RT-PCR for the presence of the virus. Hemolymph is thought to be a common dissemination route from the gut to the salivary glands for many circulative viruses. In the present study, WBNV could not be detected in the hemolymph of any of the life stages, suggesting that the main route of WBNV to infect the PSG is not through the hemolymph. These findings are consistent with earlier reports by Ullman, Nagata, and colleagues [38,39]. Tospovirus particles have not been detected in the hemolymph of thrips in any study. A potential barrier to tospovirus presence in the hemolymph of thrips has not been identified to date. The role of the TSG in disseminating tospoviruses from the midgut to the PSG [9,40] was not supported in the present study, as infection signal was observed in the PSG before the TSG was infected. Instead, transport of WBNV from the midgut to the PSG was primarily through thread-like connecting ligaments. The TSG may have an ancillary role in virus dissemination in later developmental stages. 

A further question that arises is why adult thrips cannot acquire and transmit tospoviruses. We did not observe any physical barrier that might hypothetically restrict WBNV movement to the PSG during the adult stage as proposed by Nagata and colleagues [7] for *F. occidentalis* and TSWV. In our study, the connections of the PSG to the midgut through ligaments and the TSG were unbroken in *T. palmi* adults. We suggest two potential scenarios: First, WBNV may not have had sufficient time after acquisition by adult thrips to infect the midgut, reach the PSG, and build up to a sufficient titer for infection of a host plant during feeding. Second, tospoviruses are known to infect epithelial cells of the anterior midgut during early larval stages [7,9,10,30], but may not be able to infect the epithelial cells of adult thrips [7,9,30]. This second hypothesis was supported by the present study as WBNV infection was evident in epithelial-like cells of the midgut in early larvae, but infection of epithelial cells was not observed in adult specimens. TSWV enters its vector through a receptor (50-kDa protein) on the surface of epithelial cells of *F. occidentalis* similar to other insect systems [41]. These receptors are more abundant in the midgut epithelial cells of larvae than of adults [34]. In some insects, the midgut epithelium is continuously regenerated or complete epithelium is shed and replaced immediately, whereas in other cells, replacement is rare [42]. It is currently unknown what changes occur in the gut epithelial cells of *T. palmi* during its transformation from larva to adult and its impact on WBNV infection or replication. The midgut epithelium may also undergo apoptosis either due to virus infection or normal development [30], resulting in virus-free conditions of the newly formed epithelium. 

In conclusion, there are both similarities and differences in the route of dissemination of WBNV in its vector, *T. palmi* compared to the well-studied TSWV-*F. occidentalis* pathosystem. The anterior midgut of *T. palmi* was the first tissue to be infected by WBNV, and it reached the PSG primarily via connecting ligaments. The TSG may also have an ancillary role in the dissemination of WBNV in the older instars of its vector, *T. palmi*. The present study contributes to the understanding of thrips–tospovirus interactions and may help to identify novel targets for better managing thrips and thrips-borne viruses. 

## Figures and Tables

**Figure 1 cells-10-00392-f001:**
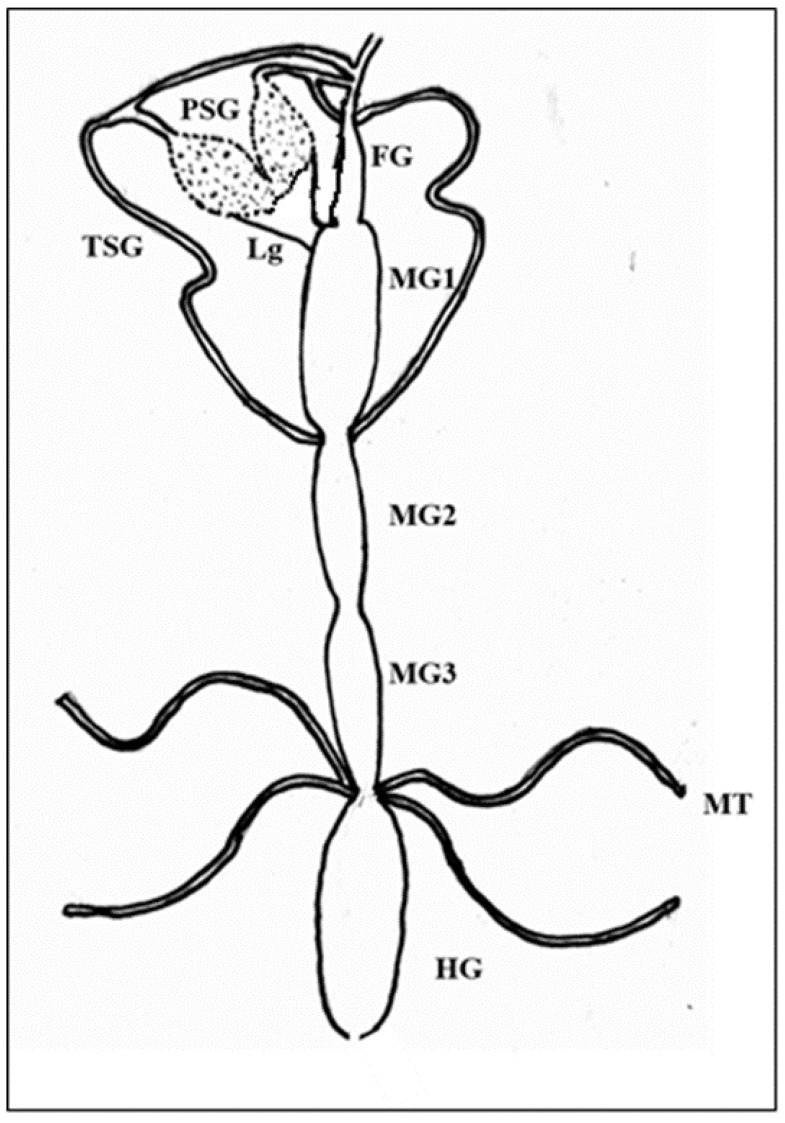
Schematic diagram of alimentary canal and salivary glands of *Thrips palmi.* FG—foregut; MG1—midgut 1; MG2—midgut 2; MG3—midgut 3; HG—hindgut; PSG—principal salivary gland; TSG—tubular salivary gland; Lg—ligament; MT—Malpighian tubules. PSG is connected to the anterior midgut by ligaments. TSG originates from the basal region of PSG and meets the alimentary canal at the junction of MG1 and MG2.

**Figure 2 cells-10-00392-f002:**
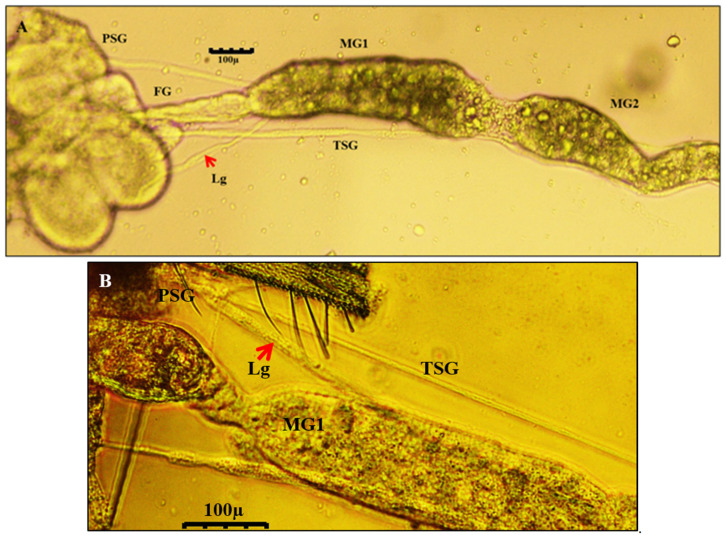
Salivary glands and alimentary tract of *Thrips palmi* viewed by light microscopy. (**A**) Larval stage (**B**) Adult stage. Figure 1. midgut 1; MG2—midgut 2; PSG—principal salivary gland; TSG—tubular salivary gland. TSG meets the midgut at the junction of MG1 and MG2. A pair of ligaments (Lg; red arrow) connects PSG with the anterior part of MG1. The connection of PSG with midgut via ligaments and TSG remains intact across the life stages.

**Figure 3 cells-10-00392-f003:**
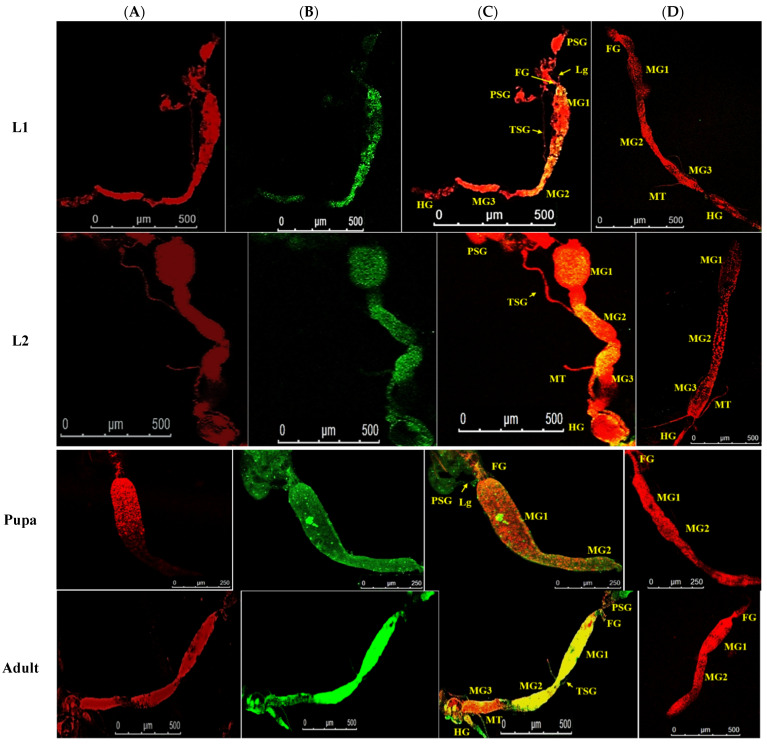
Localization of WBNV N protein in different life stages of *Thrips palmi*. Larval (L1, L2), pupal and adult stages of WBNV-exposed and non-exposed *T. palmi* were dissected. The specimens were labeled with Alexa Fluor 594 phalloidin, and rabbit antibodies against WBNV N protein followed by FITC-conjugated goat anti-rabbit antibody. Alexa Fluor 594- and FITC-specific fluorescence signals were captured by confocal microscopy. (**A**). Alexa Fluor 594 phalloidin-conjugated F-actin probe emitted red fluorescence (**B**). FITC-conjugated goat anti-rabbit antibody emitted green fluorescence indicating the presence of WBNV N protein in WBNV-exposed *T. palmi* (**C**). Merging both Alexa Fluor 594 and FITC signals yielded yellow-orange shades in parts of the image where both red and green fluorescence were emitted (**D**). Merged Alexa Fluor 594 and FITC-specific fluorescence signals in non-exposed *T. palmi* indicating absence of WBNV. Specimens belonging to different life stages of *T. palmi* are indicated on the left. FG—foregut; MG1—midgut 1; MG2—midgut 2; MG3—midgut 3; HG—hindgut; PSG—principal salivary gland; TSG—tubular salivary gland; Lg—ligament; MT—Malpighian tubules. In L1, the infection of WBNV was indicated by presence of green fluorescence in the anterior midgut 8 h post acquisition. A low intensity green fluorescence was also recorded in PSG and ligaments connecting gut to PSG. However, TSG was not infected by WBNV, indicated by the absence of the green fluorescence signal. Infection of WBNV spread to other parts of the alimentary canal during L2 48 h post acquisition. Multiplication of WBNV in PSG and anterior midgut during pupal and adult stages was evident from an increased level of N protein green fluorescence.

**Figure 4 cells-10-00392-f004:**
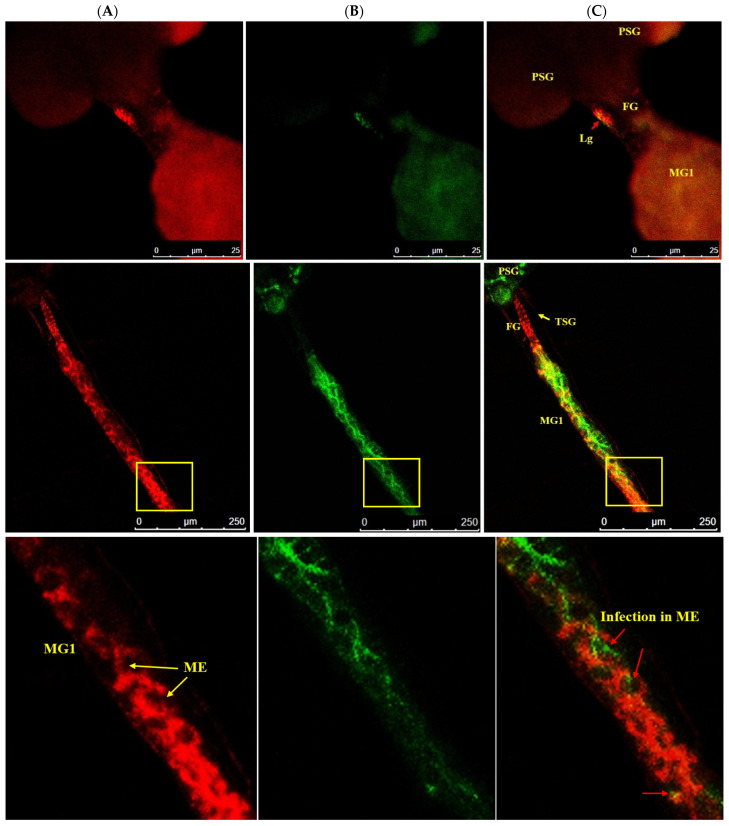
Accumulation of WBNV in ligaments of *T. palmi* during L1. (**A**). Alexa Fluor 594 phalloidin-conjugated F-actin probe red fluorescence (**B**). FITC-conjugated goat anti-rabbit secondary antibody green fluorescence indicating WBNV-N protein (**C**). Merged image of red and green fluorescence signals. FG—foregut; MG1—midgut 1; ME—epithelial-like cells of midgut (yellow arrows, bottom panel); Lg—ligament (red arrow, top panel); PSG—principal salivary gland; TSG—tubular salivary gland (yellow arrow, middle panel). WBNV infection was detected in ligaments connecting MG1 to PSG 12 h post acquisition (top panel). TSG was not infected by WBNV during the L1 stage indicated by the absence of the green fluorescence signal (middle panel). Panels marked with yellow boxes are enlarged in the bottom panel. Epithelial-like cells in MG1 are marked by yellow arrows (bottom panel). Infection of WBNV (red arrows) in epithelial-like cells of MG1 was seen in L1 12 h post acquisition.

**Figure 5 cells-10-00392-f005:**
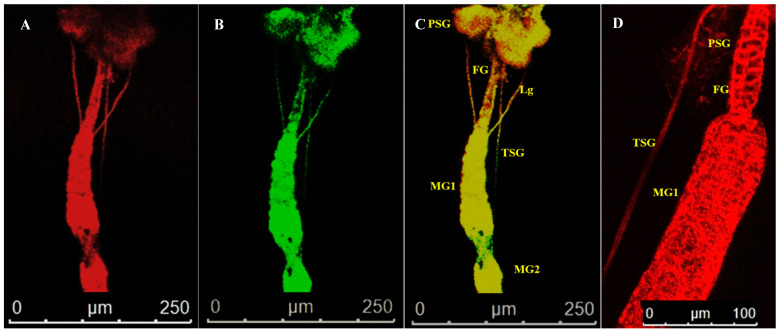
Accumulation of WBNV in ligaments and TSG of *Thrips palmi* during L2. (**A**). Alexa Fluor 594 phalloidin-conjugated F-actin probe emitting red fluorescence (**B**). FITC-conjugated goat anti-rabbit secondary antibody yielded green fluorescence indicating the presence of WBNV-N protein (**C**). Merging both Alexa Fluor 594 and FITC fluorescence signals yielded yellow-orange shades. (**D**). Merged Alexa Fluor 594 and FITC-specific fluorescence signals in non-exposed *T. palmi* indicating absence of WBNV. FG—foregut; MG1—midgut 1; MG2—midgut 2; PSG—principal salivary gland; TSG—tubular salivary gland; Lg—ligament. WBNV infection progressed from anterior midgut to PSG via ligaments 60 h post acquisition. Infection of TSG was also visible in some specimens during L2 stage.

**Figure 6 cells-10-00392-f006:**
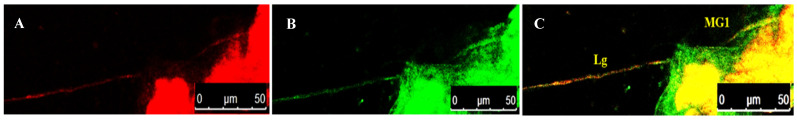
Accumulation of WBNV in ligaments connecting PSG to anterior midgut during P2 stage of *Thrips palmi*. (**A**). Alexa Fluor 594 phalloidin-conjugated F-actin probe red fluorescence (**B**). FITC-conjugated goat anti-rabbit secondary antibody green fluorescence indicating WBNV-N protein (**C**). Merged image of red and green fluorescence signals. MG1—midgut 1; Lg—ligament. Ligaments connecting midgut to PSG were intact during pupal stages. WBNV infection was detected in ligaments connecting MG1 to PSG.

**Figure 7 cells-10-00392-f007:**
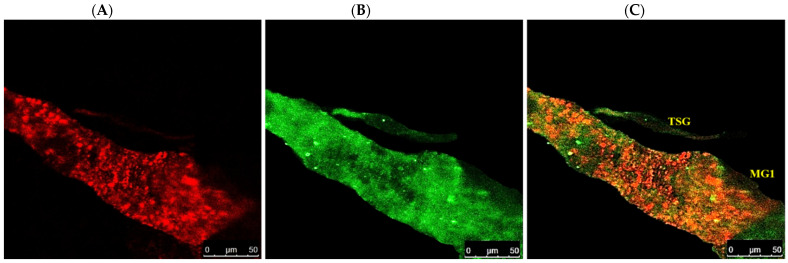
Accumulation of WBNV in TSG and alimentary canal during P2 stage of *Thrips palmi*. (**A**). Alexa Fluor 594 phalloidin-conjugated F-actin probe red fluorescence (**B**). FITC-conjugated goat anti-rabbit secondary antibody green fluorescence indicating WBNV-N protein (**C**). Merged image of red and green fluorescence signals. MG1—midgut 1; TSG—tubular salivary gland. TSG meets the alimentary canal at the junction of MG1 and MG2. WBNV infection in TSG was prominent in P2 8 days post acquisition.

**Figure 8 cells-10-00392-f008:**
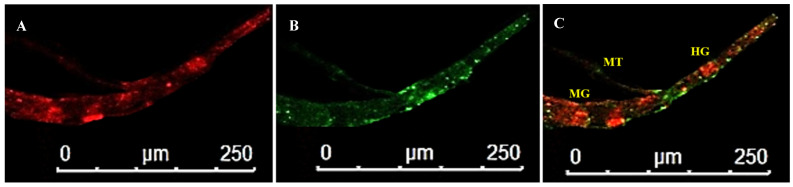
WBNV accumulation in hindgut and Malpighian tubules during P2 stage of *Thrips palmi***.** (**A**). Alexa Fluor 594 phalloidin-conjugated F-actin probe red fluorescence (**B**). FITC-conjugated goat anti-rabbit secondary antibody green fluorescence indicating WBNV-N protein (**C**). Merged image of red and green fluorescence signals. HG—hindgut; MG—midgut; and MT—Malpighian tubules. WBNV infection was distinguished by FITC green fluorescence in hindgut and Malpighian tubules in P2, 8 days post acquisition.

**Figure 9 cells-10-00392-f009:**
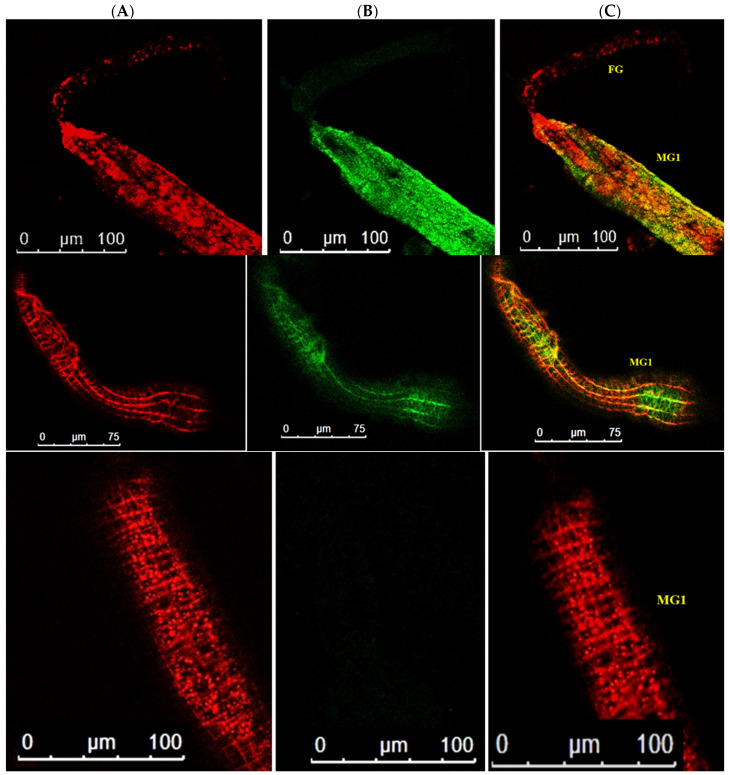
Accumulation of WBNV in anterior midgut of *Thrips palmi* adult. (**A**). Alexa Fluor 594 phalloidin-conjugated F-actin probe red fluorescence (**B**). FITC-conjugated goat anti-rabbit secondary antibody green fluorescence indicating WBNV-N protein (**C**). Merged image of red and green fluorescence signals. FG—foregut; MG1—midgut 1. Level of WBNV infection was highest in anterior midgut (top panel). Green fluorescence indicating WBNV N protein increased in midgut 12 days post acquisition. Infection of longitudinal and circular muscle cells of the midgut was evident by presence of green fluorescence (middle panel). No FITC green fluorescence was observed in the midgut of aviruliferous *T. palmi* (bottom panel).

**Figure 10 cells-10-00392-f010:**
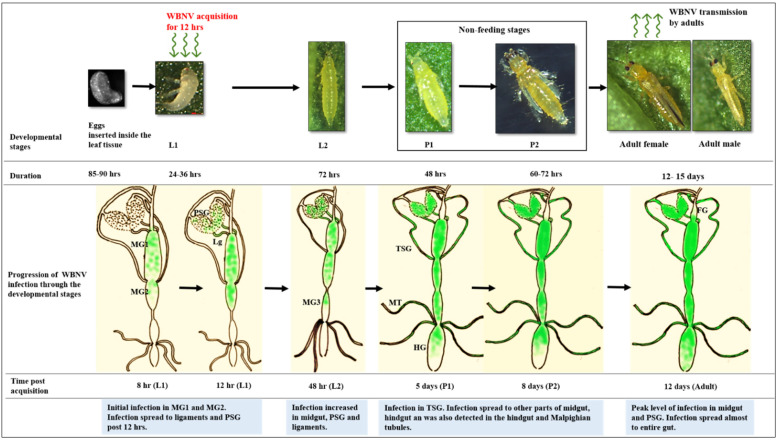
A schematic representation of WBNV dynamics from egg to adult stages of *Thrips palmi*. FG—foregut; MG1—midgut 1; MG2—midgut 2; MG3—midgut 3; HG—hindgut; PSG—principal salivary gland; TSG—tubular salivary gland; Lg—ligament; MT—Malpighian tubules. Microscopic kidney shaped eggs are inserted inside the leaf tissue under natural conditions. First instar larvae (L1) were exposed to WBNV for 12 h. The first infection of WBNV was observed in anterior midgut comprising MG1 and MG2. Infection spread to PSG from midgut through connecting ligaments 12 h post acquisition during late L1 stage. Infection increased in MG1, MG2, anterior portion of MG3, PSG, and ligaments in second instar larvae (L2) 48 h post acquisition. Infection in TSG was prominent in all samples during pupal stages (P1, P2). WBNV infection was also detected in the hindgut and Malpighian tubules 8 days post acquisition. Peak level of infection in PSG and midgut was observed during adult stage. Infection spread almost through the entire gut 12 days post acquisition.

## Data Availability

The sequence data generated in this study are available in NCBI with accession numbers MT992047 and MW080513.

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
