# Peer review of "Progression of Watermelon Bud Necrosis Virus Infection in Its Vector, Thrips palmi"

_cells, 2021, doi:10.3390/cells10020392_

Round 1
Reviewer 1 Report
Bunyavirales includes more than 4800 species viruses with single-stranded negative-sense RNA genomes. As a prototype of the plant-infected Tospoviridae family, Tomato spotted wilt orthotospovirus (TSWV) is well known to be transmitted mainly by Frankliniella occidentalis Pergande (Western flower thrips, WFT) in a persistent and propagative manner. The progression route of TSWV in F. occidentalis has been the subject of significant research and the general pathway of the virus in the vector begins with feeding on the plant host and uptake of virus particles. The virus travels through the alimentary canal and the first tissue infected is the anterior region of the midgut (MG). Amalendu Ghosh et al. report new data that virus progression in another insect system melon thrips (Thrips palmi) infected by watermelon bud necrosis virus (WBNV). These data are still at a pretty preliminary stage. Here are major concerns for this manuscript.
- No quantitative viral data was provided. Most of data are immune-staining with WBNV-nucleocapsid protein antibodies detected with FITC-conjugated secondary antibodies. They are descriptive. We have no idea about whether specific kind of cell type could support WBNV replicates.
2. The authors mainly report the one route of WBNV in vector thrips. No data about another common hemolymph route which is thought to be a major way for many other circulative viruses to travel from the gut to the salivary glands. They may also provide new evidences to indicate that WBNV release and movement in the hemolymph is required for infection of the salivary glands.
- Too much strong background signal with Actin antibody for all figures except Fig. 7 and 8. Seems like these signal not only masks green signal which indicates the presence of virus, but also make the cell shape unclear.
- Please refer the method of the immunofluorescence microscopy used for virus-insect interaction described in Zheng Limin et al., Virus Gene “Infection route of rice grassy stunt virus, a tenuivirus, in the body of its brown planthopper vector, Nilaparvata lugens (Hemiptera: Delphacidae) after ingestion of virus”.
- Many of technical detail information not provide. Below are some examples of areas where the manuscript is difficult to follow. However, I emphasize that these are exemplary for revisions that need to be made throughout the manuscript.
Minor concern:
- How many days of L2 instar of the palm thrips? When they did this assay for L2 samples. Please clearly present these important parameters.
- Line 185. Virus infection influence vector's survival, the question of which stage thripes' survival was affected? More detail data should be given to support your conclusion.
- how many days “after acquisition of virus”?line 241“ level”means what?
- “Faint green fluorescence”. maybe its the background fluorescence of the tissues, have you ever do the negtive control? the figure should be shown.
- Line 256 figure 3. Maybe too much Alexa Fluor 594 phalloidin was used during the staining, actin and cell profile is not clear because of exceedingly red fluorescence.
- Line 274 figure4. Does virus locate in the cells of MG1 or in the lumen?
- Figure 5. This figure is not clear. Do you mean all the green part is infection by virus?
- Figure 9 Green fluorescence merged with red on the Transverse Muscle Longitudinal Muscle of gut. The red fluorescence is OK because it labeled the actin protein. But the green fluorescence seems like a background fluorescence.
Author Response
Response to reviewer’s comments:
We are thankful to the Reviewers and Academic Editor for critically going through the MS and suggesting substantial improvements. All the suggestions made by the reviewers and editor have been incorporated into the revised MS. Our response to each comment is shown below. The changes made in the manuscript are indicated in red colour in the text. We trust that our revisions will make this MS acceptable for publication in the journal Cells.
Reviewers’ comments:
Reviewer #1:
Bunyavirales includes more than 4800 species viruses with single-stranded negative-sense RNA genomes. As a prototype of the plant-infected Tospoviridae family, Tomato spotted wilt orthotospovirus (TSWV) is well known to be transmitted mainly by Frankliniella occidentalis Pergande (Western flower thrips, WFT) in a persistent and propagative manner. The progression route of TSWV in F. occidentalis has been the subject of significant research and the general pathway of the virus in the vector begins with feeding on the plant host and uptake of virus particles. The virus travels through the alimentary canal and the first tissue infected is the anterior region of the midgut (MG). Amalendu Ghosh et al. report new data that virus progression in another insect system melon thrips (Thrips palmi) infected by watermelon bud necrosis virus (WBNV). These data are still at a pretty preliminary stage. Here are major concerns for this manuscript.
Comment 1: No quantitative viral data was provided. Most of data are immune-staining with WBNV-nucleocapsid protein antibodies detected with FITC-conjugated secondary antibodies. They are descriptive. We have no idea about whether specific kind of cell type could support WBNV replicates.
Response: This is the first such study in a new virus-vector system, therefore this represents a largely descriptive report. We don’t have any evidence whether a specific cell type supports WBNV replication. We have mentioned this in line no 479-481. This question is worthy of further investigation. Identification of cell types involved in replication of WBNV is beyond the scope of this MS.
Comment 2: The authors mainly report the one route of WBNV in vector thrips. No data about another common hemolymph route which is thought to be a major way for many other circulative viruses to travel from the gut to the salivary glands. They may also provide new evidences to indicate that WBNV release and movement in the hemolymph is required for infection of the salivary glands.
Response: We agree with the reviewer that hemolymph is a common route for circulation of virus from midgut to salivary gland in most of the circulative transmissions. In case of thrips-tospovirus, it is evident that tospoviruses are not circulated through the hemolymph (Rotenberg et al. Current Opinion in Virology. 2015; Ullman et al. Phytopathology. 1995). However, the potential barrier to tospovirus circulation in thrips hemolymph is uncharacterized so far. In the present study, we have tested thrips hemolymph, but WBNV was not detected. Considering this previously reported evidence in thrips-tospovirus relationships, we did not mention the negative results in our original manuscript. As per suggestion by the reviewer, we have now mentioned the absence of WBNV in the hemolymph in the revised manuscript (line no. 46-49; 117-118; 410-411; 523-533). Our findings are consistent with Ullman et al. Phytopathology. 1995 and Nagata T. Ph.D. thesis. Agricultural University of Wageningen, Wageningen, the Netherlands. 1999.
Comment 3: Too much strong background signal with Actin antibody for all figures except Fig. 7 and 8. Seems like these signal not only masks green signal which indicates the presence of virus, but also make the cell shape unclear.
Response: We have tried to improve the figures in question by reducing red (actin) fluorescence signal. A separate panel (panel B) is also provided where only the virus-labeled FITC green signals (no actin red signal) is captured.
Comment 4: Please refer the method of the immunofluorescence microscopy used for virus-insect interaction described in Zheng Limin et al., Virus Gene “Infection route of rice grassy stunt virus, a tenuivirus, in the body of its brown planthopper vector, Nilaparvata lugens (Hemiptera: Delphacidae) after ingestion of virus”.
Response: referred as suggested
Comment 5: Many of technical detail information not provide. Below are some examples of areas where the manuscript is difficult to follow. However, I emphasize that these are exemplary for revisions that need to be made throughout the manuscript.
Response: Further important technical details have been added to the revised manuscript.
Minor concern:
Comment 6: How many days of L2 instar of the palm thrips? When they did this assay for L2 samples. Please clearly present these important parameters.
Response: Mentioned in the revised ms (line no. 190, 316)
Comment 7: Line 185. Virus infection influence vector's survival, the question of which stage thripes' survival was affected? More detail data should be given to support your conclusion.
Response: Survival was lower during emergence of adults from P2 stage. The effect of WBNV on survival of thrips is not the focus of the present manuscript. The details of WBNV effect on T. palmi are available in our previous publication. A summary of that has now been included in Discussion. (Ghosh A, Basavaraj YB, Jangra S, Das A. Exposure to watermelon bud necrosis virus and groundnut bud necrosis virus alters the life history traits of their vector, Thrips palmi (Thysanoptera: Thripidae). Archives of Virology. 2019; 164(11):2799–804).
Comment 8: how many days “after acquisition of virus”?line 241“ level”means what?
Response: additional information provided in the text as suggested.
Comment 9: “Faint green fluorescence”. maybe its the background fluorescence of the tissues, have you ever do the negtive control? the figure should be shown.
Response: We respectfully disagree. Negative control was done for each set of experiments. Representative images of negative controls were shown in Fig 10 of previous version of manuscript. Background fluorescence has not been evident anywhere. As suggested, images of respective individual negative controls have now been included in revised Figure 3.
Comment 10: Line 256 figure 3. Maybe too much Alexa Fluor 594 phalloidin was used during the staining, actin and cell profile is not clear because of exceedingly red fluorescence.
Response: Twelve (16 in revised manuscript after including control) images were accommodated in a single frame in Fig 3. To cover the entire gut and salivary glands, the images were captured in lower magnification. In low magnification, cell profile was not clear and florescence signals appeared brighter. It was not possible to cover the entire gut as well as to show the cell profile in a single image. High magnification images of important parts have been shown in other figures. As suggested, we have tried to reduce the red signal evenly in all images as far as possible.
Comment 11: Line 274 figure 4. Does virus locate in the cells of MG1 or in the lumen?
Response: Virus infected the cells of midgut. In Fig 4, the upper panel shows infection in the midgut, no infection in TSG. Bottom panel shows infection in ligament marked by red arrow.
Comment 12: Figure 5. This figure is not clear. Do you mean all the green part is infection by virus?
Response: Figure 5 shows PSG and anterior midgut section only, where maximum virus-labeled fluorescence was observed.
Comment 13: Figure 9 Green fluorescence merged with red on the Transverse Muscle Longitudinal Muscle of gut. The red fluorescence is OK because it labeled the actin protein. But the green fluorescence seems like a background fluorescence.
Response: We respectfully disagree with the reviewer. We have included the negative control image for this set that clearly shows that this is not background fluorescence. Localization of WBNV was distinguished in horizontal and circular muscle tissues of midgut. Our findings are consistent with Han et al. Frontiers in Microbiology. 2019; Nagata. Ph.D. thesis. Agricultural University of Wageningen, 1999. This is mentioned in Results and Discussion of the revised ms (line no. 405-407; 516-518)
Reviewer 2 Report
Manuscript cells-1045997 describes the movement of watermelon bud necrosis virus in the thrips vector, Thrips palmi. The research is of high interest and thoroughly conducted. In addition, the manuscript is very well written. Nonetheless, a few minor edits could be considered for improvement. See recommendations below:
Lines 69 and 95: Change brinjal to eggplant
Line 116: ... and salivary glands were carefully removed from the rest of the thrips body and placed ...
Lines 173 and 173: It would be nice to provide GenBank accession numbers for a few sequences showing 100% and 99.99% nucleotide sequence identity for the mtCOI and WBNV-N sequences that were determined in this study, respectively.
Line 176: Change above to in Materials and methods
Line 199: ... in the image. The level of fluoresence varied ...
Line 243: ... fluorescence indicating no WBNV presence. A low intensity ...
Line 249: Change indicate to indicated
Line 294: Eliminate was
Author Response
Response to reviewer’s comments:
We are thankful to the Reviewers and Academic Editor for critically going through the MS and suggesting substantial improvements. All the suggestions made by the reviewers and editor have been incorporated into the revised MS. Our response to each comment is shown below. The changes made in the manuscript are indicated in red colour in the text. We trust that our revisions will make this MS acceptable for publication in the journal Cells.
Reviewer’s comments
Reviewer #2:
Manuscript cells-1045997 describes the movement of watermelon bud necrosis virus in the thrips vector, Thrips palmi. The research is of high interest and thoroughly conducted. In addition, the manuscript is very well written. Nonetheless, a few minor edits could be considered for improvement. See recommendations below:
Comment 1: Lines 69 and 95: Change brinjal to eggplant
Response: done
Comment 2: Line 116: ... and salivary glands were carefully removed from the rest of the thrips body and placed ...
Response: revised as suggested
Comment 3: Lines 173 and 173: It would be nice to provide GenBank accession numbers for a few sequences showing 100% and 99.99% nucleotide sequence identity for the mtCOI and WBNV-N sequences that were determined in this study, respectively.
Response: GenBank accessions of sequences that are identical with studied organisms have been mentioned in the revised MS.
Comment 4: Line 176: Change above to in Materials and methods
Response: revised as suggested
Comment 5: Line 199: ... in the image. The level of fluoresence varied ...
Response: revised as suggested
Comment 6: Line 243: ... fluorescence indicating no WBNV presence. A low intensity ...
Response: revised as suggested
Comment 7: Line 249: Change indicate to indicated
Response: changed as suggested
Comment 8: Line 294: Eliminate was
Response: changed as suggested
Round 2
Reviewer 1 Report
I don't think the authors have solved critical concerns raised on the first version of ms. I suggest they can perform some experiments suggested to solve some of them instead of just explaining this is out of scope of this research.
Author Response
Epithelial-like cells appear to be involved in the replication of WBNV in L1 stage (Fig. 4) 12 hrs post acquisition. Infection in muscle tissues was recorded in older instar (Fig 9). An in-depth study of cell types involved may be done in future to shed further light on replication and cellular movement of this virus.